# Vasoplegic Syndrome after Cardiopulmonary Bypass in Cardiovascular Surgery: Pathophysiology and Management in Critical Care

**DOI:** 10.3390/jcm11216407

**Published:** 2022-10-29

**Authors:** Zied Ltaief, Nawfel Ben-Hamouda, Valentina Rancati, Ziyad Gunga, Carlo Marcucci, Matthias Kirsch, Lucas Liaudet

**Affiliations:** 1Service of Adult Intensive Care, Lausanne University Hospital and University of Lausanne, 1010 Lausanne, Switzerland; 2Service of Anesthesiology, Lausanne University Hospital and University of Lausanne, 1010 Lausanne, Switzerland; 3Service of Cardiac Surgery, Lausanne University Hospital and University of Lausanne, 1010 Lausanne, Switzerland

**Keywords:** vasoplegic syndrome, cardio-pulmonary bypass, cardiac surgery, vasopressin, angiotensin 2, methylene blue, hydroxocobalamin, nitric oxide

## Abstract

Vasoplegic syndrome (VS) is a common complication following cardiovascular surgery with cardiopulmonary bypass (CPB), and its incidence varies from 5 to 44%. It is defined as a distributive form of shock due to a significant drop in vascular resistance after CPB. Risk factors of VS include heart failure with low ejection fraction, renal failure, pre-operative use of angiotensin-converting enzyme inhibitors, prolonged aortic cross-clamp and left ventricular assist device surgery. The pathophysiology of VS after CPB is multi-factorial. Surgical trauma, exposure to the elements of the CPB circuit and ischemia-reperfusion promote a systemic inflammatory response with the release of cytokines (IL-1β, IL-6, IL-8, and TNF-α) with vasodilating properties, both direct and indirect through the expression of inducible nitric oxide (NO) synthase. The resulting increase in NO production fosters a decrease in vascular resistance and a reduced responsiveness to vasopressor agents. Further mechanisms of vasodilation include the lowering of plasma vasopressin, the desensitization of adrenergic receptors, and the activation of ATP-dependent potassium (K_ATP_) channels. Patients developing VS experience more complications and have increased mortality. Management includes primarily fluid resuscitation and conventional vasopressors (catecholamines and vasopressin), while alternative vasopressors (angiotensin 2, methylene blue, hydroxocobalamin) and anti-inflammatory strategies (corticosteroids) may be used as a rescue therapy in deteriorating patients, albeit with insufficient evidence to provide any strong recommendation. In this review, we present an update of the pathophysiological mechanisms of vasoplegic syndrome complicating CPB and discuss available therapeutic options.

## 1. Introduction

Vasoplegic syndrome (VS) complicating cardiopulmonary bypass (CPB) is a frequently described syndrome with many used denominations: low vascular resistance syndrome, catecholamine refractory vasoplegia, cardiac vasoplegia syndrome, inflammatory response to bypass, systemic inflammatory response syndrome (SIRS) after cardiac surgery, post perfusion syndrome and vasoplegic shock post-bypass [1,2,3,4,5,6]. The clinical pattern is that of a distributive form of circulatory shock developing in the first 24 h after CPB [7], characterized by hypotension (mean arterial pressure < 65 mmHg resistant to fluid challenge), systemic vascular resistance < 800 dynes s/cm^5^ and a cardiac index > than 2.2 l/min/m^2^ [1,8].

The incidence of VS after CPB varies from 5 to 44% [1,8] and it accounts for 4.6% of all forms of circulatory shock [9]. A higher incidence has been reported in patients with low preoperative left ventricular (LV) ejection fraction (EF) and in those undergoing LV assist device surgery [10,11]. Several predisposing factors have been identified, such as advanced age, prolonged aortic cross-clamp and CPB time, as well as the pre-operative use of angiotensin-converting enzyme inhibitors (ACEI) and diuretics [8,10,12]. Patients experiencing VS display an increased incidence of postoperative complications and, consequently, a longer intensive care unit (ICU) stay [8], as well as greater post-operative mortality (5.6–15%) [2,13] than in the overall cardiac surgery population (3.4–6.2%) [14,15]. Table 1 summarizes the main clinical characteristics, pathophysiological mechanisms, and principles of management of VS after CPB.

## 2. Pathophysiology of Vasoplegic Syndrome after CPB (Figure 1)


### 2.1. Physiology of Vascular Smooth Muscle Cell Contraction

The phosphorylation of the regulatory Myosin Light Chain (MLC) is the cornerstone of smooth muscle contraction. Vasoconstricting agents such as norepinephrine, angiotensin 2 and vasopressin activate cell surface G-protein-coupled receptors (GPCRs) to induce an increase in intracellular Ca^++^ concentration in vascular smooth muscle cells (VSMCs). In turn, Ca^++^ binds to calmodulin, forming a complex activating MLC Kinase to promote MLC phosphorylation, with subsequent actin-myosin interaction and smooth muscle contraction [16]. In contrast, vasodilators such as nitric oxide (NO) and natriuretic peptides trigger a guanylyl cyclase-dependent increase in cyclic guanosine monophosphate (cGMP), which activates myosin phosphatase to dephoshorylate MLC, thereby preventing smooth muscle contraction [17,18]. CPB promotes a series of pathophysiological mechanisms concurring to alter smooth muscle Ca^++^ signaling and MLC phosphorylation, which may ultimately result in a state of vasoplegia, characterized by pathological vasodilation and reduced responsiveness to vasoconstrictors.

### 2.2. Inflammatory Pathways Triggered by CPB

Several mechanisms triggered at the time of CPB, and amplified upon its discontinuation, concur to trigger a systemic inflammatory response: the exposure of blood components to the artificial surfaces of the extracorporeal circuit, surgical trauma, organ ischemia, the release of endotoxin from the gut into the circulation, and hemolysis with liberation of free hemoglobin. Altogether, these mechanisms promote the activation of the complement cascade and the expression of pro-inflammatory mediators, such as interleukin-1 beta (IL-1β), interleukin-6 (IL-6) and tumor necrosis factor alpha (TNFα) [6,19,20]. IL-6 is notably a potent inhibitor of vascular contraction by increasing the synthesis of cyclic AMP, which promotes vasodilation by reducing myoplasmic [Ca^++^] [21,22]. Accordingly, higher circulating levels of IL-6 have been associated with an increased incidence of VS and a higher need for vasopressors after CPB [23,24]. Longer CPB and aortic cross-clamping durations, combined surgery and redo intervention all predispose to a more intense inflammatory response and therefore represent independent risk factors for the development of VS [25]. Following the discontinuation of CPB, systemic reperfusion promotes the generation of oxygen-free radicals and the amplification of the initial inflammation. Moreover, the reinfusion of cell saver blood containing hemolyzed red blood cells, activated platelets and denatured proteins, may also contribute to this response [4,26]. Ultimately, such systemic inflammation engages a series of biological processes precipitating the loss of vascular tone, as detailed below.

### 2.3. Adrenoreceptor Desensitization

Inflammatory cytokines such as IL-1β, IL-6 and TNFα impair adrenoreceptor-dependent signaling by two mechanisms. The first one is related to a reduced expression of vascular α1-adrenoreceptors, consecutive to suppressed promoter activity at the level of gene transcription [27]. The second and most important mechanism relies in the desensitization of the receptors, triggered by the excessive release of catecholamines in response to baroreceptor and cytokine-dependent stimulation of the central sympathetic system. In turn, sustained adrenergic stimulation induces the phosphorylation of the G-protein coupled adrenoreceptor through the activation of GPCR kinases, inhibiting catecholamine binding and downstream signaling [26,28].

### 2.4. Increased Nitric Oxide Biosynthesis

Inflammatory cytokines stimulate the expression of the inducible isoform of NO synthase (iNOS) through the activation of nuclear factor kappa B (NF-κB) [29]. In contrast to the low amounts of NO released by the constitutive, calcium-dependent NOS isoforms, the calcium-independent iNOS has the ability to produce copious quantities of NO over prolonged periods of time [29]. Elevated iNOS expression has been reported in lung tissue after CPB [30], and increased generation of NO has been shown to be proportional to the duration of CPB [31].

Several mechanisms account for the vasodilatory properties of NO. First and most importantly, NO activates soluble guanylyl cyclase in VSMCs to promote the formation of cyclic GMP, leading to MLC dephosphorylation [29]. Secondly, NO (via cGMP) activates adenosine triphosphate-sensitive potassium channels (K_ATP_) in VSMCs, causing K^+^ efflux and membrane hyperpolarization. This results in the closure of voltage-activated calcium (Ca^++^) channels, the reduction in cytosolic Ca^++^ and vasodilation [32]. These effects are amplified by the ability of NO-cGMP to further promote K^+^ efflux by activating the vascular calcium-activated potassium channels (KCa^++^) [32]. An additional mechanism of NO-induced vasodilation is the generation of peroxynitrite, a strong oxidant and nitrating species formed from the rapid reaction of NO with the superoxide anion radical (O_2_^−^). Peroxynitrite contributes to vasoplegia by impairing bioenergetics in VSMCs, by activating matrix metalloproteinases, and by inativating catecholamines and their receptors [29].

### 2.5. Relative Deficiency of Vasopressin

Inflammatory cytokines and sustained baroreceptor stimulation are responsible for an overactivation of the central sympathetic system and the hypothalamo-pituitary axis, with subsequent release of high levels of norepinephrine, epinephrine, cortisol, and vasopressin [26]. Persistent shock and hypotension result in the progressive decline in the blood levels of these vasoactive mediators, a phenomenon particularly significant in the case of vasopressin [26]. Circulating vasopressin levels during CPB have been measured by Colson et al. in sixty-four consecutive patients. Patients developing VS displayed a significant drop of plasma vasopressin 8 h after CPB in comparison to non-VS patients (16 vs. 42 pmol/l, *p* = 0.01) [33]. Comparable results have been reported in patients with septic shock [34,35]. In turn, such decrease in vasopressin abrogates its vasopressor effects, which depend on the vascular V1 receptor (V1R), whose activation results in a protein kinase C (PKC)-dependent increase in intracellular calcium, both directly via Ca^++^ channel opening, and indirectly via K_ATP_ channel inhibition [36]. In addition, V1R-dependent PKC activation may inhibit MLC phosphatase, reduce iNOS expression and NO formation in response to inflammatory cytokines, promote the release of vasoconstrictors by endothelial cells (endothelin-1) and platelets (thromboxane A2), and ultimately can increase the vascular sensitivity to catecholamines [37,38,39,40].

### 2.6. Activation of K_ATP_ Channels and Membrane Hyperpolarization in VSMCs

The regulation of membrane potential in VSMCs depends on the activity of several types of K^+^ channels whose opening promotes K^+^ efflux and membrane hyperpolarization, resulting in the inhibition of Ca^++^-channel-dependent cellular influx of Ca^++^ and subsequent vasodilation. These channels include voltage dependent (Kv), Ca^++^ activated (KCa), inward rectifier (Kir) and ATP-sensitive (K_ATP_) K^+^ channels [41]. Among these channels, the activation of K_ATP_ channel has been closely linked to the development of pathological vasodilation and the induction of hypotension in various forms of distributive shock [17]. Several mechanisms may account for the activation of K_ATP_ channel in VS associated with CPB, including NO release, vasopressin deficiency, hypoxia, acidosis and an increase in the generation of hydrogen sulfide (see below) [17,36,42,43,44,45].

### 2.7. Dysfunction of the Renin-Angiotensin System

Angiotensinogen, an inactive peptide produced by the liver, is cleaved by renin into angiotensin 1 (Ang1), and then to angiotensin 2 (Ang2) by the action of type 1 angiotensin-converting enzyme (ACE 1) in the lung endothelium. In turn, Ang2 targets the vascular AT1 receptor, resulting in increased cytosolic Ca^++^ and vasoconstriction. Under conditions of impaired ACE 1 activity (pulmonary dysfunction), Ang2 formation is prevented, and Ang1 is converted into the vasodilating derivative Ang_1–7_ by the type 2 ACE (ACE 2) [46,47]. Such scenario might develop during the pulmonary exclusion associated with CPB, precipitating vasoplegia by reducing ACE 1 activity and Ang2 formation, while increasing Ang1 and Ang_1–7_. In addition, this process may be amplified by the enhanced secretion of renin, triggered in response to low Ang2 and reduced blood pressure, which further increases Ang1 formation (“high renin shock”) [47].

### 2.8. Endothelial Glycocalyx Alteration

The glycocalyx is a complex layer of glycosaminoglycans and proteoglycans coating the endothelial cell surface, with important components comprising heparan sulfate and syndecan-1. Heparan sulfate plays an important role in the regulation of vascular tone by modulating shear-stress dependent NO production, whereas syndecan-1 has been associated with anti-inflammatory effects by downregulating leukocyte adhesion and activation at the surface of the endothelium [48,49]. Circumstantial evidence indicates that CPB may induce damage to the glycocalyx, as reported in a pilot study by Boer et al., who found that glycocalyx thickness was significantly reduced after the initiation of CPB, a change which persisted after weaning and was associated with microcirculatory impairment [50]. Furthermore, Abou-Arab et al. reported that the plasma levels of syndecan-1 increased after CPB, consistent with glycocalyx shedding. Interestingly, patients developing VS had reduced baseline plasma syndecan-1 and a lesser increase after CPB, suggesting that altered glycocalyx structure with reduced syndecan-1 might predispose to endothelial inflammation and vasodilation after CPB [51].

### 2.9. Possible Role of an Excess Production of Hydrogen Sulfide

Hydrogen sulfide (H_2_S) is a gaseous transmitter formed in the vascular system from the metabolism of cysteine and homocysteine by the enzymes cystathionine-gamma-lyase (CSE), cystathionine-beta-synthase (CBS), and 3-mercaptopyruvate sulfurtransferase (3-MST) [52]. In blood vessels, H_2_S plays several homeostatic functions, notably acting as a vasodilator. This effect is related, on one hand, to the activation of K_ATP_ channels leading to membrane hyperpolarization [53], and on the other hand, to the enhancement, via multiple mechanisms, of vascular NO signaling [52]. An increased formation of H_2_S has been demonstrated in various animal models of septic shock, which might participate to the development of pathological vasodilation in such conditions [54]. Although not directly demonstrated in the setting of CPB, one can assume that the prevailing inflammatory conditions might enhance the synthesis of H_2_S as reported in sepsis. This would be consistent with the notion that hydroxocobalamin, which can bind H_2_S and inhibit its effects, can increase blood pressure in patients with severe VS post-CPB, as discussed later [55,56].

## 3. Predisposing Factors of Post-CPB Vasoplegic Syndrome

### 3.1. Patient-Related Factors

Several pre-operative conditions have been associated with an increased incidence of VS post-CPB, including advanced age, recent myocardial infarction, anemia, and higher perioperative risk score (Euro score) [25,57,58]. Heart failure with reduced ejection fraction is of particular concern, as indicated by an increased incidence of VS up to 74% in patients with poor LV function (EF < 37%) exposed to prolonged CPB duration [23]. In a prospective cohort of 145 cardiac surgery patients, Argenziano et al. reported that an EF < 35% was an independent risk factor for post-CPB VS, with a relative risk of 9.1 [10]. Additionally, a low EF (<40%) has been correlated with the development of severe systemic inflammation after CPB, as diagnosed by circulating levels of IL-6 > 1000 pg/mL [10,24]. In line with these notions, patients undergoing heart transplantation [59] or LVAD surgery [60] are at increased risk of VS. Pre-operative renal failure is another important risk factor, possibly due to higher levels of pro-inflammatory cytokines and vascular endothelial dysfunction in these patients [61]. In a meta-analysis including > 30,000 patients from 10 studies, Dayan et al. reported that pre-operative renal failure was indeed the most significant independent risk factor for VS, with an odds ratio of 1.47 (95% CI 1.17–1.86) [25].

### 3.2. Pre- and Peri-Operative Drug Therapies

The peri-operative use of ACE inhibitors (ACEI) has been reported as an independent risk factor for VS after CPB in several observational studies [62], a concept substantiated by the results of a recent meta-analysis of 10 observational case–control studies [63]. Although these findings have led to the frequent practice to withdraw ACEI prior to cardiac surgery, this issue remains highly controversial, as no association between ACEI and VS was found in two randomized (but underpowered) controlled trials (RCTs) [64,65], and in the previously mentioned meta-analysis by Dayan et al. [25]. The latter study pointed out the large heterogeneity between studies (type and dose of ACEI used, definition of post-CPB VS), making a clear statement impossible. Therefore, the precise link between ACEI and VS after CPB remains uncertain, which underscores the need for large prospective RCTs to address this issue.

The peri-operative use of diuretics is a further risk factor to consider, as emphasized in a retrospective analysis on 1992 patients showing an independent association between diuretics and VS, with an odds ratio (OR) of 1.36 (1.07–2.38) [57]. Reduced cardiac preload, as well as hyponatremia-dependent blunting of vasopressin secretion, could explain, at least partly, this observation [57]. In contrast to ACEI and diuretics, treatment with beta-blockers has been associated with a reduced incidence of VS in the meta-analysis by Dayan et al. [25]. By preventing chronic stimulation of the sympathetic system, beta-blockers might improve the sensitivity of adrenergic receptors and reduce the development of vasoplegia after CPB [66].

A significant association between the preoperative use of inotropes and postoperative vasoplegia has also been reported, primarily related to the use of sympathomimetic inotropes (dobutamine), but not phosphodiesterase inhibitors (milrinone) [57]. Conversely, milrinone may reduce the incidence of VS, as shown in patients undergoing cardiac transplantation, through yet undefined mechanisms [67,68].

## 4. Outcome

Few studies reported data about the outcome of patients experiencing VS. Compared to non-complicated postoperative course, patients with VS require higher inotropic support, display more frequent bleeding complications, and have more organ dysfunction, including liver injury, renal and respiratory failure, as well as more frequent lactic acidosis [8,25,33,69], translating into a prolonged duration of mechanical ventilation and length of ICU stay [12,25]. Ultimately, such complications have a profound impact on survival, with reported mortality rates between 5.6 and 15%, as compared to 3.4 to 6.2% in the overall population of cardiac surgery patients [2,13,14,15]. The ominous prognosis of VS has been particularly underscored by Levin et al. in a retrospective cohort of 2823 adult cardiac surgery patients. Those developing VS after the discontinuation of CPB were significantly more likely to have a prolonged ICU stay length > 10 days or to die in hospital, with an OR of 3.3 (*p* = 0.005) [12]. This was further substantiated by the very high mortality (25%) reported in heart transplant recipients with post-operative VS, contrasting to 9% mortality in those without VS [68].

## 5. Management

### 5.1. Peri-Operative Prevention

Patient care during the peri-operative period is crucial to reduce the risk of VS after CPB. In this context, Van Vessem et al. proposed a risk score incorporating age, sex, the type of surgery, the value of creatinine clearance and thyroxine levels, the presence of anemia and the use of beta-blockers [58]. Using this score, the authors stratified the risk of developing VS into low, intermediate, and high risk, with an observed incidence of VS of 13, 39 and 65%, respectively. They suggested that, according to the calculated score, preoperative measures such as hemodynamic optimization and improvement of renal function could prove beneficial to reduce the risk of post-operative VS. This score should now be validated in a prospective multicenter trial.

Given the lack of consensus regarding pre-operative drug management (use of diuretics, ACEI and beta-blockers), an individualized strategy should be implemented on a case-by-case basis. With respect to surgery, beyond the obvious requirement to keep the duration of CPB and aortic cross-clamping to a minimum, it has been suggested that the use of minimal extracorporeal circulation (MECC) and of biocompatible, heparin-coated short circuits, might limit inflammation by reducing the exposure to foreign surfaces [26]. In an evidenced-based review of 98 RCTs evaluating various strategies to modulate inflammatory response during CPB, Clive Landis et al. identified 8 RCTs examining MECC (median sample size = 45) and 14 RCTs evaluating biocompatible circuits (median sample size = 38). Although several trials reported a reduction in inflammation, the overall clinical benefits were limited [70]. In a meta-analysis of 24 RCTs comparing MECC with conventional ECC, Anastasiadis et al. reported that MECC was associated with significantly less systematic inflammation, a reduced incidence of VS (OR 0.19, 95% CI 0.04–0.88, *p* = 0.03) and decreased mortality [71]. These results must however be interpreted with caution, owing to the small number of included patients in most studies evaluated and the heterogeneity of outcome measurements. Therefore, doubt persists regarding the real impact of these strategies, and additional, adequately powered RCTs are warranted to address their effectiveness.

### 5.2. Volume Resuscitation

The main goal of resuscitation in any form of circulatory shock is to ensure adequate tissue perfusion and oxygen delivery to meet the tissue metabolic needs. Such requirement implies the maintenance of sufficient cardiac output and perfusion pressure [72]. Fluid resuscitation is crucial to maintain adequate preload and optimize cardiac output. Both absolute (pre- and peri-operative fluid losses) and relative (venous vasodilation with increased unstressed vascular volume) hypovolemia [73] may reduce venous return and cardiac preload in the cardiac surgery patient. Although it is critical to recognize and correct hypovolemia, avoiding fluid over-resuscitation is of paramount importance given the harmful consequences of increased extravascular lung water, excessive cardiac filling pressure and hemodilution [74,75,76]. Volume resuscitation should therefore be based on the demonstration of insufficient perfusion, which can be clinically evaluated by the mottling score and the capillary refill time [77], and biologically by the value of arterial blood lactate and the venous-to-arterial difference in the partial pressure of carbon dioxide (VA-pCO2 gap) [78].

It is generally admitted that beyond an initial fluid loading of 20–30 mL/kg, additional fluids should be cautiously administered [79], and guided by dynamic indices (e.g., pulse pressure variation, echographic indices of stroke volume variation) to confirm a positive effect of fluid loading on cardiac output [75,80]. The technique of “mini fluid challenge” (administration of 100 to 150 mL crystalloid fluid over 60 to 120 s) may be here interesting [81]. This was recently shown in a multicenter prospective study in surgical patients undergoing laparotomy surgery [82], indicating that such challenge predicted fluid responsiveness, defined by an increase in stroke volume index ≥ 10%, with high-sensitivity and specificity.

Cardiac surgical patients frequently display reduced hemoglobin concentrations postoperatively, which could increase the risk of tissue hypoxia. Several RCTs therefore evaluated transfusions thresholds in this population. Although one RCT (TITRe2 study) found reduced 90 d mortality (secondary outcome) when patients were transfused according to a liberal (Hb 9 g/dL) vs. restrictive (Hb 7.5 g/dL) threshold, most RCTs and meta-analyses indicate non-inferiority of restrictive strategies with respect to morbidity and mortality [83,84,85,86]. Hence, current guidelines recommend a transfusion threshold of 7.5 g/dL in stable patients without evidence of tissue hypoxia [87,88]. However, no study evaluated transfusion thresholds in the specific population of VS patients. It is noteworthy that anemia can reduce systemic vascular resistance, both by reducing blood viscosity and decreasing Hb-dependent inhibition of NO [89,90], which could contribute to aggravate hypotension in VS patients. Indeed, a direct relationship exists between Hb levels and blood pressure [91]. Therefore, anemia may be less well-tolerated in post-operative patients with VS, and transfusion thresholds might be higher in this setting, which should be evaluated in future studies. For these reasons, we use a 9 g/dL transfusion threshold in VS patients treated in our center.

### 5.3. Vasoactive Drugs

A mean arterial pressure (MAP) of 65–70 mm Hg is considered as the initial value to target in circulatory shock, as higher values have not been associated with improved survival [75,92,93]. After reaching such MAP, the appropriate target must be repeatedly assessed, using the clinical and biochemical endpoints measures of tissue perfusion previously mentioned [94]. Although it has long been considered that vasopressors should be initiated following adequate volume resuscitation, it is now recommended to start vasoactive drugs together with volume resuscitation, as this strategy has been associated with reduced short-term mortality in sepsis-associated vasoplegia [95]. Potential benefits from early vasopressor use include a reduced need of fluids, via the mobilization of blood from the unstressed and stressed vascular compartments, and an improved myocardial perfusion through an increased diastolic pressure [96].

#### 5.3.1. Norepinephrine

Norepinephrine (NE) targets the vascular *α*-1 adrenoreceptor to increase intracellular Ca^++^ and promote vascular smooth muscle contraction. Although no mortality benefit from its use in post-CPB VS has been demonstrated, NE remains the first line vasopressor agent in most cardiac surgery centers, and is thus still considered as the standard of care [97,98,99,100]. However, owing to its beta-adrenergic actions [97,98,99,100], NE may trigger important side effects such as tachycardia, atrial fibrillation, increased myocardial oxygen consumption, and hyperlactatemia [97,98,99,100]. Such detrimental effects are even more common with epinephrine and dopamine, which therefore are not recommended in the therapy of VS [97,98,99,100]. It is also noteworthy that high doses NE may result in immunosuppression predisposing to secondary infections [101]. Therefore, accumulating evidence indicates that sympathetic overactivation should be avoided, and that a non-catecholaminergic vasopressor (vasopressin) should be added early if MAP cannot be rapidly restored with NE, or in case of side effects attributable to NE [97,98,99,100]. There is currently no recommendation regarding the threshold dose of NE for the initiation of vasopressin, as such threshold varied between 0.1 and 0.7 μg/kg/min across studies evaluating this issue [97,98,99,100].

#### 5.3.2. Vasopressin

As previously mentioned, vasopressin (VP) promotes vasoconstriction via vascular V1-receptor-dependent increase in cytosolic Ca^++^, modulation of NO signaling, and improvement of catecholamine sensitivity. The reduced circulating levels of VP reported in patients with VS after CPB (see above) provides a strong rationale for the therapeutic use of exogenous VP in this setting. This was initially evaluated in a small prospective RCT published in 1998 by Argenziano et al. [10]. Ten vasoplegic hypotensive patients following LVAD surgery and treated with NE received VP (0.1 U/min) or saline placebo. In all patients, VP significantly increased MAP while allowing a reduction in NE requirements, an effect that was particularly marked in a subgroup of patients displaying inappropriately low circulating levels of endogenous VP. Several subsequent studies in patients with vasodilatory shock confirmed similar findings, further reporting a reduced incidence of new onset arrhythmias with VP in comparison to NE [102,103]. Additionally, various authors reported that the systemic vasopressor effect of VP occurred in the absence of any negative influence on pulmonary vascular resistance, thereby increasing right ventricular perfusion pressure without increasing right ventricular afterload [104,105].

A large RCT (VANCS trial), including 300 patients with VS after cardiac surgery, directly compared NE (10–60 μg/min) with VP (0.01–0.06 U/min) as first line vasopressor to maintain a MAP of 65 mm Hg [13]. Patients in the VP arm had a significant reduction (32% vs. 49%, *p* = 0.0014) of the primary outcome (a composite of 30 days mortality or severe complications), mostly due to a marked reduction in acute renal failure (10% vs. 36%, *p* < 0.0001). In addition, VP was associated with a lower incidence of atrial fibrillation (64% vs. 82%, *p* = 0.0004), and did not result in a greater occurrence of digital, mesenteric, or myocardial ischemia.

Based on these data, and on meta-analyses confirming a reduced incidence of AF and possible decreased incidence of acute kidney injury with VP [106,107,108], a recent expert consensus [97] proposed the following recommendations. (1) To start or add VP to increase MAP in case of adverse effects related to sympathoadrenergic drugs (strong recommendation, moderate level of evidence). (2) To use VP as a first line vasopressor therapy (weak recommendation, moderate level of evidence). Furthermore, owing to the favorable profile of action of VP on the pulmonary circulation, this expert consensus recommended the use of VP (first line or added to norepinephrine) in cardiac surgical patients with right ventricle dysfunction and/or pulmonary hypertension (weak recommendation, very low level of evidence). Regarding the dose of VP, no definitive consensus has emerged so far, but due to a dose-dependent increased risk of ischemic complications, doses higher than 0.06 U/min should be avoided.

#### 5.3.3. Angiotensin 2

The possibility that a disturbed renin angiotensin system leading to reduced Ang2 generation participates to VS after CPB (see above) suggests that exogenous Ang2 might be a therapeutic option in this setting. Several case reports and small case series reported safe and successful administration of Ang2, with significant hemodynamic improvement and vasopressor sparing effect, in vasoplegic patients following cardiac surgery (see [47] for an extensive recent review on this topics). The ATHOS-3 trial evaluated the effectiveness of Ang2 (20–200 ng/kg/min) in refractory vasodilatory shock from various origins (primarily septic shock) unresponsive to high dose of vasopressors [109]. The pre-defined hemodynamic target (MAP increase of at least 10 mm Hg or an increase to at least 75 mm Hg after 3 h) was reached significantly more often with Ang2 than placebo (70 vs. 23%, OR 7.95, 95% CI 4.76–13.3, *p* < 0.001). A subsequent post hoc analysis in 16 patients with VS after cardiac surgery showed that the end point was reached by 89% of patients treated with Ang2, compared to 0% in those receiving the placebo, an effect achieved with very low doses of Ang2 (5 ng/kg/min) in a majority of patients. In addition, there was a marked vasopressor sparing effect of Ang2, with a 76.5% decrease in vasopressors, compared to a 7.8% increase in the placebo group (*p* = 0.0013) [110]. Taken together, these findings strongly suggest that Ang2 may provide significant benefits in VS after CPB, and some have indeed already incorporated Ang2 in their treatment algorithm [47]. However, current evidence remains insufficient to make any recommendation [97], implying the need for additional RCTs evaluating Ang2 in cardiac surgery patients [111].

#### 5.3.4. Methylene Blue

Methylene blue (MB) is a thiazine dye used as an antidote to treat methemoglobinemia [112]. Owing to several pharmacological actions, MB increases vascular tone, a property that led to evaluate MB for the therapy of vasoplegia associated with septic shock or following CPB. MB acts by inhibiting NO-dependent vasodilation via three distinct mechanisms: direct scavenging of NO, inhibition of NO synthase, and most significantly, inhibition of guanylyl cyclase [97,98,99,100]. Several investigations in cardiac surgery patients (reviewed in [113].

In spite of these encouraging results, data on clinical outcome with MB remain scarce and contrasted. In a RCT comparing pre-operative MB (2 mg/kg over 30 min) vs. placebo in patients at high risk of post-operative VS, Özal et al. reported a significant lower incidence of VS (0% vs. 26%, *p* < 0.001), as well as lower ICU and hospital stay in the MB arm [114]. Levin et al. randomized 56 patients with established post-operative VS to receive either MB (1.5 mg/kg) or placebo, and reported a significant reduction in mortality with MB (0% vs. 21.4%, *p* = 0.01) [115,116,117]. In a retrospective analysis of 221 patients with per-operative vasoplegia treated with MB (2 mg/kg), Kofler et al. reported improved hemodynamic status, but unchanged 90-day mortality [113]. In contrast to these data, Weiner et al. found, in a retrospective cohort of 226 patients with post-operative VS, that treatment with MB (57 patients) was an independent predictor of in-hospital mortality (*p* = 0.007) and post-operative complications (*p* = 0.001). In addition, Mehaffey et al. showed that MB treatment, in a cohort of 118 patients, was associated with a high rate of post-operative complications and mortality [118]. It must also be underscored that MB may be associated with serious complications, including serotoninergic syndrome, acute hemolysis in patients with G6PDH deficiency, and impaired splanchnic perfusion at high doses (7 mg/kg).

Therefore, given the lack of high-quality data, uncertainties regarding clinical outcomes, the potential for severe side effects, unresolved issues regarding the timing (pre-, per-, or post-operatively), dose and mode of administration (bolus vs. infusion), the use of MB to treat VS after CPB is presently not recommended, unless as a rescue therapy in cases with hypotension refractory to usual vasopressors [97].

#### 5.3.5. Hydroxocobalamin

Hydroxocobalamin (vitamin B12) is used for the therapy of pernicious anemia and in cyanide poisoning [119]. Several properties of hydroxocobalamin indicate that it may increase vascular tone, including inhibition of NOS enzymes [120], direct NO inactivation [121] and reduction in H_2_S toxicity through direct binding [122,123]. Hydroxocobalamin has been therefore evaluated for the treatment of refractory VS after CPB, using the same protocol of administration as in cyanide poisoning (5 g administered by IV infusion over 15 min). Several case reports showed that such regimen increased MAP and allowed reduction in vasopressors [124], and comparable effects have been reported in small case series [125,126]. In a retrospective study of 33 patients treated with hydroxocobalamin for refractory hypotension during or after CPB, a pressor effect was noted in 24 patients, albeit such response was highly heterogeneous [127]. Indeed, four distinct patterns were identified, including no response (“poor responders”, 27%), brisk and sustained increase (“responders”, 24%), progressive increase (“sustainers”, 27%) and brisk increase followed by rebound hypotension (“rebounders”, 21%) [127]. Since such heterogeneity might be due, partly, to the mode of administration (bolus), Seelhammer et al. evaluated the effects of a continuous infusion of hydroxocobalamin (5 g over 6 h) post-operatively in 12 patients with severe VS, showing that such regimen permitted a prolonged (>10 h) and significant reduction in vasopressors in all patients [128].

To sum up current available data, hydroxocobalamin appears generally associated with hemodynamic improvement and decreased vasopressor requirement in the setting of VS after cardiac surgery, as reviewed recently by Shapeton et al. [124]. However, safety and mortality data are lacking, and both the timing (pre- or post-operative) and mode (bolus vs. continuous infusion) of administration remain to be established. Therefore, the use of hydroxocobalamin should only be considered as a rescue strategy in deteriorating patients refractory to usual vasopressors.

#### 5.3.6. Vitamin C

Vitamin C (ascorbic acid) is a co-factor for several enzymes involved in the biosynthesis of endogenous catecholamines [129], and it also increases the sensitivity of adrenoreceptors [130]. It is also a free radical scavenger and might therefore reduce oxidant-mediated tissue injury, inflammation and endothelial dysfunction [131]. An important reduction in plasma Vitamin C occurs following CPB, suggesting a potential therapeutic role from exogenous supplementation [132]. This was first assessed by Wieruszewski et al. in 3 patients with severe VS after cardiac surgery. High dose Vitamin C (1500 mg iv every 6 h) allowed to rapidly reduce vasopressors in all patients. Such approach was then evaluated in a RCT including 50 cardiac surgery patients with VS [133]. Although Vitamin C was generally well tolerated, it did not allow a statistically faster resolution of vasoplegia. It is also worth to mention that a recent study in septic shock patients (LOVIT trial, 872 patients) did not find any benefit from Vitamin C treatment (50 mg/kg q6 h for up to 96 h) [134]. In contrast, patients treated with Vitamin C displayed significantly higher risk of death or persistent organ dysfunction. Additionally, in another trial in septic shock, Vitamin C was associated with a higher need for renal replacement therapy and greater positive fluid balance [135]. Altogether, the results of these trials do not support the use of vitamin C to treat vasodilatory shock, and therefore, it cannot be recommended in the setting of VS after CPB.

### 5.4. Anti-Inflammatory Strategies

#### 5.4.1. Corticosteroids

The interest of corticosteroids in the treatment of vasoplegia involves two main pharmacological actions. The first one is related to their anti-inflammatory effects, leading notably to a reduced expression of inflammatory cytokines, iNOS and cyclooxygenase-2 [97,99,136]. The second one relies in the ability of corticosteroids to enhance the synthesis of catecholamines and to increase the expression and sensitivity of adrenoreceptors [137,138]. In septic shock, the administration of low doses corticosteroids (hydrocortisone) accelerates the reduction in vasopressors and the resolution of shock, albeit with controversial effects on mortality, as shown in several major RCTs [139,140,141,142]. In cardiac surgery, two large trials (SIRS trial -7507 patients, and DECS trial—4494 patients) evaluated the effects of high doses corticosteroids (intraoperative methyprednisolone or dexamethasone) on mortality and major complications, and did not report any significant effects of the interventions [143,144]. With respect to low doses hydrocortisone, several trials (extensively reviewed in [145]) reported reduced inflammation, lower incidence of atrial fibrillation, shortened ICU and hospital length of stay and reduced need of vasopressors. However, none of the studies specifically addressed the role of low doses steroids in established VS after cardiac surgery. Therefore, although low doses steroids appear safe and may be associated with some beneficial effects, there is no evidence supporting their use to specifically treat CPB-induced vasoplegia.

#### 5.4.2. Extracorporeal Cytokine Adsorption Therapy

Extracorporeal cytokine adsorption therapy (ECAT) is a technique of extracorporeal blood purification using specifically designed filters able to adsorb and remove inflammatory mediators from the circulation. This strategy has been applied to decrease inflammation in sepsis, with promising results reported in a number of case reports and case series. However, prospective RCTs failed to report any mortality benefit, some of them showing instead potential harmful effects possibly related to the unselective removal of anti-inflammatory mediators and drugs, notably antibiotics (see: [146] for extensive review on this topic).

In the field of cardiac surgery, the largest RCT to date has been published by Diab et al., who compared ECAT using the CytoSorb^®^ cytokine filter integrated to the CPB circuit (*n* = 142 patients) with standard of care (*n* = 146 patients). Although patients in the intervention arm displayed significantly lower levels of IL-1*β* and IL-18 at the end of CPB, they did not differ from controls in terms of the primary outcome (post-operative change from baseline in sequential organ failure assessment score), notably with respect to the use of vasopressors [147]. In a systematic review including 5 RCTs (*n* = 163 patients) evaluating ECAT in cardiac surgery, Goetz et al. reported no significant benefits from the technique in terms of mortality and post-operative complications [148]. Moreover, a recent meta-analysis of 8 trials in sepsis and 10 in cardiac surgery (total 875 patients), revealed (low certainty of evidence) that the use of a cytokine filter might increase mortality in critically ill patients with inflammatory conditions [149]. As a whole, these data indicate that the efficacy and safety of ECAT remains not established. Therefore, its application in cardiac surgical patients cannot be currently recommended.

### 5.5. Management of VS after CPB: Summary of Evidence and Current Recommendations (Table 1)

The aim of the management of VS is to restore organ perfusion pressure and adequate oxygen delivery, by ensuring appropriate preload and by the administration of vasoactive drugs, with the aim to maintain a MAP of 65 mm Hg. Among the panoply of available agents, present evidence indicates that conventional vasopressors are recommended as first line therapy [97]. In this respect, norepinephrine is generally considered as the standard of care. Vasopressin should be added to norepinephrine in case of untoward side effects (tachycardia, atrial fibrillation) related to excessive sympathetic stimulation, or could be used as the initial vasopressor, as supported by the recent VANCS clinical trial [13]. Non-conventional vasopressors, including Methylene Blue, Angiotensin 2 and Hydroxocobalamin, have been increasingly used in refractory cases, but evidence is not sufficient to make any recommendations. Therefore, while awaiting further RCTs to evaluate their efficacy and safety, these drugs should only be introduced as a rescue therapy, on a case-by-case basis. Low doses hydrocortisone may be associated with vasopressors in order to potentiate their effects, due to their demonstrated role in shortening the duration of vasoplegia in sepsis and their safety profile. However, evidence for their use in post-CPB VS has not yet been demonstrated. Finally, in spite of some theoretical advantages, Vitamin C and cytokine adsorption filters should not be used, owing to possible deleterious effects.

## 6. Conclusions and Perspectives

Vasoplegic syndrome is a frequent complication of cardiopulmonary bypass, with significant impact on major clinical outcomes. It is of particular concern in patients with left ventricular dysfunction or/and chronic renal failure, and it occurs more frequently after complex surgery, left ventricular device implantation and heart transplantation. Systemic inflammation, triggered by multiple mechanisms set in motion during CPB and after its discontinuation, is the key pathophysiological event leading to vasoplegia, by promoting a series of biological processes impairing the normal maintenance of vascular tone. Beyond the obvious necessity to shorten, as much as possible, the duration of CPB to prevent or reduce such deleterious consequences, therapy of established VS is based primarily on adequate volume resuscitation and the use of vasoactive agents to maintain end-organ perfusion and viability. It is puzzling that, in spite of the importance of this conundrum, only very few and adequately powered controlled prospective studies have addressed the question of the best regimen of vasoactive drugs to apply in this setting. Therefore, the current recommended strategy relies only on low to moderate evidence indicating that conventional vasopressors, which include norepinephrine and vasopressin, either alone or in combination, should be used to restore vascular tone, whereas non-conventional vasopressors (methylene blue, hydroxocobalamin and angiotensin 2) are not recommended, except from extremely refractory situations.

While future studies will undoubtedly help optimize treatment algorithms, it must be underscored that vasoplegic syndrome develops as a final common disorder precipitated by an array of pathophysiological mechanisms. Such mechanistic diversity might hamper the development of a unique therapeutic strategy similarly applicable to every single patient (“one size fits all”). We may therefore hope that future developments in the field of molecular diagnostics and precision medicine might allow decipher which mechanism prevails in a given situation, and thereby tailor the therapeutic management to each specific patient.

## Figures and Tables

**Figure 1 jcm-11-06407-f001:**
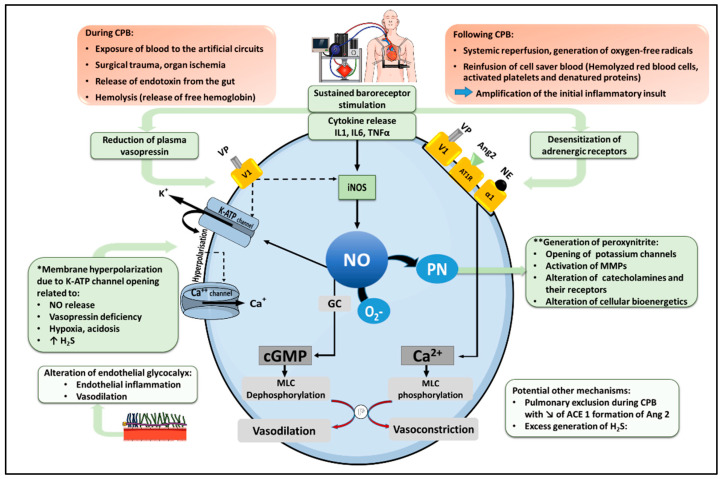
Pathophysiological mechanism of vasoplegia after cardio-pulmonary bypass. Under normal conditions, norepinephrine (NE), angiotensin 2 (Ang2) and vasopressin (VP) activate their respective receptors in vascular smooth muscle cells, namely the α1-adrenoreceptor, the Ang2 type I receptor (AT1R) and the vasopressin V1 receptor, to promote an increase in cytosolic Ca^++^, leading to Myosin Light Chain (MLC) phosphorylation, actin-myosin interaction and smooth muscle contraction. In addition, vasopressin exerts indirect vasoconstrictor actions by inhibiting two physiological mechanisms of vasodilation, including NO production and the opening of ATP-dependent potassium channels (K_ATP_). Following cardiopulmonary bypass (CPB), cytokine release and sustained baroreceptor stimulation foster a down-regulation and desensitization of adrenergic receptors, as well as a reduction in plasma VP, reducing catecholamine and vasopressin-mediated vasoconstriction. In addition, inflammatory cytokines stimulate the expression of the inducible isoform of NO synthase (iNOS), leading to excessive NO production. NO activates soluble guanylyl cyclase (GC), generating cyclic GMP (cGMP), which dephosphorylates MLC with subsequent vasodilation. Moreover, NO activates K_ATP_ channels, causing K^+^ efflux, membrane hyperpolarization, and the closure of Ca^++^ channels, thereby reduce cytosolic Ca^++^. K_ATP_ channels opening and membrane hyperpolarization can also result from additional mechanisms, as indicated (*). NO further reacts with the superoxide anion radical (O_2_^−^) to form peroxynitrite (PN), which contributes to vasodilation via multiple processes, as indicated (**). CPB can also trigger alterations of the glycocalyx (favoring endothelial dysfunction and vascular inflammation), promote the production of hydrogen sulfide (H_2_S) with vasodilating properties, and reduce Ang2 formation by type I angiotensin-converting enzyme (ACE 1). Solid lines indicate activation; dashed lines indicate inhibition.

**Table 1 jcm-11-06407-t001:** Vasoplegic shock after cardiopulmonary bypass—main characteristics.

Definition	Distributive Form of Circulatory Shock ≤ 24 h after CPB Initiation, Characterized by: (1) MAP < 65 mmHg resistant to fluid challenge (2) SVR < 800 dynes s/cm^5^ (3) CI > 2.2 L/min/m^2^ *
Predisposing factors	
*Patient-related factors*	Advanced age; anemia; low LVEF; renal failure
*Pre-/peri-operative drugs*	Diuretics; sympatho-adrenergic inotropes; ACEI (controversial)
*Operative factors*	CPB/aortic cross clamping time; redo surgery; combined surgery; LVAD surgery; HTx
Pathophysiology	
*Initiating events*	Systemic inflammatory response triggered by: (1) Exposure of blood to the artificial surfaces of the extra-corporeal circuit; (2) Surgical trauma; (3) Ischemia/ reperfusion injury; (4) Oxidative stress; (5) Release of endotoxin from the gut; (6) Hemolysis; (7) Reinfusion of cell saver blood
*Mechanisms of pathological vasodilation*	(1) Desensitization of adrenergic receptors; (2) Increased NO biosynthesis; (3) Low plasma vasopressin; (4) VSMC hyperpolarization due to opening of K_ATP_ channels; (5) RAS dysfunction with low Ang2 (?); (6) Excess H_2_S generation (?); (7) Endothelial glycocalyx alteration (?).
Outcome	(1) More frequent postoperative bleeding; (2) Increased incidence of organ dysfunction: renal, liver and respiratory failure; (3) Prolonged mechanical ventilation and length of ICU/hospital stay; (4) Higher mortality.
Management	I. Ensure adequate cardiac preload and cardiac output (1) Evaluate tissue perfusion by capillary refill time, mottling score, arterial lactate, VA-PCO_2_ gap; (2) Use dynamic indices of volume responsiveness. II. Vasoactive drugs (1) Conventional Vasopressors **: Norepinephrine; Vasopressin; (2) Non-conventional Vasopressors *** (in refractory VS): Methylene Blue; Hydroxocobalamin; Angiotensin 2. III. Low doses of hydrocortisone ****

Abbreviations: ACEI: Angiotensin-Converting Enzyme Inhibitors; Ang 2: Angiotensin 2; CI: Cardiac Index; CPB: CardioPulmonary Bypass; H2S: Hydrogen Sulfide; HTx: Heart Transplantation; ICU: Intensive Care Unit; MAP: Mean Arterial Pressure; LVAD: Left Ventricular Assist Device; LVEF: Left Ventricle Ejection Fraction; NO: Nitric Oxide; RAS: Renin−Angiotensin System; SVR: Systemic Vascular Resistance; VA−PCO2 gap: Venous to Arterial Difference in the Partial Pressure of Carbon Dioxide; VS: Vasoplegic Syndrome; VSMC: Vascular Smooth Muscle Cell. *: In some patients with severe cardiac dysfunction, vasoplegia may be present in spite of a low cardiac output (CI < 2.2 L/min/m^2^). These patients therefore do not fulfill the complete hemodynamic criteria of pure VS, and should be considered as presenting a mixed form of shock. **: Strong recommendation ***: Insufficient evidence; risk of harm incompletely documented; can be used on a case−by−case basis as a rescue therapy; need for additional adequately powered randomized controlled trials (RCTs). ****: Acts via anti−inflammatory actions + increased vascular responsiveness to vasoconstrictors; accelerates resolution of shock in sepsis, controversial effects on mortality; precise role in VS post−CPB presently undefined; may be used as adjunctive treatment to conventional vasopressors in refractory cases; need for additional adequately powered RCTs.

## Data Availability

Not applicable.

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
