# Peer review of "Vasoplegic Syndrome after Cardiopulmonary Bypass in Cardiovascular Surgery: Pathophysiology and Management in Critical Care"

_jcm, 2022, doi:10.3390/jcm11216407_

Round 1

Reviewer 1 Report

The authors present a review on vasoplegia following cardiopulmonary bypass. The manuscript is well written and adequate reference to existing literature in the field has been made.

However, the authors are encouraged to further differentiate and discuss the following points:

1-    In patients with severely reduced LVEF the cardiac index is most likely lower than 2.2 as given in the definition (e.g. Tab.1), the authors should attribute this sub-cohort in the definition.

2-    Please add information on recommended hematocrite in patients with vasoplegic syndrome after CPB and discuss transfusion trigger.

Author Response

Comments and Suggestions for Authors

The authors present a review on vasoplegia following cardiopulmonary bypass. The manuscript is well written and adequate reference to existing literature in the field has been made.

However, the authors are encouraged to further differentiate and discuss the following points:

Comment 1. In patients with severely reduced LVEF the cardiac index is most likely lower than 2.2 as given in the definition (e.g. Tab.1), the authors should attribute this sub-cohort in the definition.

Answer. We agree with this comment. We therefore added a footnote to Table 1 (lines 65-67), as follows: “In some patients with severe cardiac dysfunction, vasoplegia may be present in spite of a low cardiac output (CI < 2.2 l/min/m2). These patients therefore do not fulfill the complete hemodynamic criteria of pure VS, and should be considered as presenting a mixed form of shock”.

Comment 2. Please add information on recommended hematocrite in patients with vasoplegic syndrome after CPB and discuss transfusion trigger.

Answer. We thank the reviewer for this important issue, which was not covered in our original submission. We therefore added a specific paragraph on transfusion in our revised manuscript (lines 346-361), as follows:

“Cardiac surgical patients frequently display reduced hemoglobin concentrations postoperatively, which could increase the risk of tissue hypoxia. Several RCTs therefore evaluated transfusions thresholds in this population. Although one RCT (TITRe2 study) found reduced 90 d mortality (secondary outcome) when patients were transfused according to a liberal (Hb 9 g/dL) vs restrictive (Hb 7.5g/dL) threshold, most RCTs and meta-analyses indicate non-inferiority of restrictive strategies with respect to morbidity and mortality[83-86]. Hence, current guidelines recommend a transfusion threshold of 7.5g/dL in stable patients without evidence of tissue hypoxia [87, 88]. However, no study evaluated transfusion thresholds in the specific population of VS patients. It is noteworthy that anemia can reduce systemic vascular resistance, both by reducing blood viscosity and decreasing Hb-dependent inhibition of NO [89, 90], which could contribute to aggravate hypotension in VS patients. Indeed, a direct relationship exists between Hb levels and blood pressure [91]. Therefore, anemia may be less well-tolerated in post-operative patients with VS, and transfusion thresholds might be higher in this setting, which should be evaluated in future studies. For these reasons, we use a 9 g/dL transfusion threshold in VS patients treated in our center”.

Reviewer 2 Report

The topic of the paper is important and clinically relevant (the VS Incidence varies from 5% up to 44% and it accounts for 4.6% of all forms of circulatory shock).

 The structure of the paper is well done.

 The main questions in the field are covered.

 The paper describes in detail VS syndrome both its clinical pattern and its pathophysiological mechanisms. The authors gave a clear definition of VS, and its main clinical characteristics and explained the predisposing factors.

 In this paper authors carefully described and discussed the latest findings on the pathophysiological mechanisms of VS and the importance of surgical trauma, exposure to the elements of the CPB and ischemia-reperfusion trauma, which promote a systemic inflammatory response and the release of cytokines with vasodilating properties.

 The authors discussed in depth the pathophysiological mechanisms behind the VS such as: the physiology of vascular smooth muscle cell contraction, inflammatory pathways triggered by CPB, adrenoreceptor desensitization, increased Nitric Oxide biosynthesis, relative deficiency of vasopressin, activation of KATP channels and membrane hyperpolarization in VSMCs, dysfunction of the renin-angiotensin system, endothelial glycocalyx alteration and possible role of excess production of hydrogen sulfide.

 The literature on which these facts are presented and discussed is up–to–date, carefully selected, and relevant.

 This peer review article provides clinicians with important information that can help in better understanding the origin of VS as well as the best therapeutic approach.

 The article is very informative and provides very well-documented information that can be very helpful in understanding and treating patients with Vasoplegic Syndrome (VS) after cardiopulmonary bypass (CPB).

Author Response

Comments and Suggestions for Authors

The topic of the paper is important and clinically relevant (the VS Incidence varies from 5% up to 44% and it accounts for 4.6% of all forms of circulatory shock).

The structure of the paper is well done.

The main questions in the field are covered.

The paper describes in detail VS syndrome both its clinical pattern and its pathophysiological mechanisms. The authors gave a clear definition of VS, and its main clinical characteristics and explained the predisposing factors.

In this paper authors carefully described and discussed the latest findings on the pathophysiological mechanisms of VS and the importance of surgical trauma, exposure to the elements of the CPB and ischemia-reperfusion trauma, which promote a systemic inflammatory response and the release of cytokines with vasodilating properties.

The authors discussed in depth the pathophysiological mechanisms behind the VS such as: the physiology of vascular smooth muscle cell contraction, inflammatory pathways triggered by CPB, adrenoreceptor desensitization, increased Nitric Oxide biosynthesis, relative deficiency of vasopressin, activation of KATP channels and membrane hyperpolarization in VSMCs, dysfunction of the renin-angiotensin system, endothelial glycocalyx alteration and possible role of excess production of hydrogen sulfide.

The literature on which these facts are presented and discussed is up–to–date, carefully selected, and relevant.

This peer review article provides clinicians with important information that can help in better understanding the origin of VS as well as the best therapeutic approach.

The article is very informative and provides very well-documented information that can be very helpful in understanding and treating patients with Vasoplegic Syndrome (VS) after cardiopulmonary bypass (CPB).

Answer. We are thankful to the reviewer for these positive comments on our manuscript.
